# Rapid Ventricular Pacing as a Safe Procedure for Clipping of Complex Ruptured and Unruptured Intracranial Aneurysms

**DOI:** 10.3390/jcm10225406

**Published:** 2021-11-19

**Authors:** Josefin Grabert, Stefanie Huber-Petersen, Tim Lampmann, Lars Eichhorn, Hartmut Vatter, Mark Coburn, Markus Velten, Erdem Güresir

**Affiliations:** 1Department of Anesthesiology and Intensive Care Medicine, University Hospital Bonn, 53127 Bonn, Germany; stefanie.huber-petersen@ukbonn.de (S.H.-P.); lars.eichhorn@ukbonn.de (L.E.); mark.coburn@ukbonn.de (M.C.); markus.velten@ukbonn.de (M.V.); 2Department of Neurosurgery, University Hospital Bonn, 53127 Bonn, Germany; tim.lampmann@ukbonn.de (T.L.); hartmut.vatter@ukbonn.de (H.V.); erdem.gueresir@ukbonn.de (E.G.)

**Keywords:** rapid ventricular pacing, intracranial aneurysm, subarachnoid hemorrhage, controlled hypotension

## Abstract

Surgical treatment of intracranial aneurysm requires advanced technologies to achieve optimal results. Recently, rapid ventricular pacing (RVP) has been described to be an elegant technique that facilitates clip reconstruction of complex unruptured intracranial aneurysm (uIA). However, there is also a growing need for intraoperative tools to ensure safe clip reconstruction of complex ruptured intracranial aneurysm (rIA). We conducted a retrospective analysis of 17 patients who underwent RVP during surgical reconstruction of complex aneurysms. Nine patients had uIA while eight patients underwent surgery for rIA suffering from consecutive subarachnoid hemorrhage (SAH). Hemodynamic data, critical events, laboratory results, and anesthesia-related complications were evaluated. No complications were reported concerning anesthesia induction and induction times were similar between patients exhibiting uIA or rIA (*p* = 0.08). RVP induced a significant decline of median arterial pressure (MAP) in both groups (*p* < 0.0001). However, median MAP before and after RVP was not different in both groups (uIA group: *p* = 0.27; rIA group: *p* = 0.18). Furthermore, high-sensitive Troponin T (hsTnT) levels were not increased after RVP in any group. One patient in the rIA group exhibited ventricular fibrillation and required cardiopulmonary resuscitation, but has presented with cardiac arrest due to SAH. Otherwise, no arrhythmias or complications occurred. In summary, our data suggest RVP to be feasible in surgery for ruptured intracranial aneurysms.

## 1. Introduction

The treatment of complex intracranial aneurysms has evolved over the past years. Main therapeutic options are surgical reconstruction of the aneurysm as compared to endovascular treatment (e.g., using platinum coils). Treatment decision depends on presence or absence of subarachnoid hemorrhage (SAH), patient characteristics such as age and comorbidities, as well as size, location, formation, and accessibility of the aneurysm [1]. While trials suggest that coiling of ruptured intracranial aneurysms (rIA) is less invasive and show equal occlusion rates in the short term, long-term outcome and occlusion rates seem more favorable if aneurysms are clipped microsurgically [2,3]. However, these findings usually concern non-complex ruptured aneurysms located in the anterior circulation.

Microsurgical treatment of cerebral aneurysm remains challenging and perioperative morbidity and mortality depend on aneurysm characteristics such as location, size, thrombosis, neck formation, and vessel configuration [4,5]. Various techniques including temporary clipping, suction-induced decompression, hypothermia, and adenosine-induced flow arrest have been described to decompress the aneurysm sack, supporting the preparation of the aneurysm and clip positioning with all exhibiting advantages but also disadvantages and individual risks to the patient [6,7,8].

Recently, the use of rapid ventricular pacing (RVP) to induce flow reduction for the treatment of unruptured intracranial aneurysms (uIA) has been assessed in case reports and a small case series [9,10,11,12]. The advantage of RVP for surgical clip reconstruction is based on the precise control of onset and duration of flow reduction as compared to pharmacological intervention (e.g., adenosine), while intra-aneurysmal blood can be redistributed more easily when parent vessels are not blocked, e.g., during temporary clipping. In a prospective study, Konczalla et al. reported RVP to be feasible and safe in selected patients who underwent preoperative cardiac evaluation for the treatment of non-complex uIA located both in the anterior and posterior circulation [13]. However, extensive cardiac risk assessment for RVP is not feasible prior to urgent procedures such as surgical clipping of rIA. Studies illuminating the feasibility of RVP in emergency surgery for rIA have not been undertaken. Therefore, we present a retrospective case series comparing the use of RVP in elective surgery for uIAs to its use in emergency surgery of rIAs.

## 2. Materials and Methods

After approval from the Ethics Committee of the University Hospital, Bonn, Germany, and in accordance with §15 of the Medical Association Nordrheins’ professional code of conduct, medical records of patients who underwent surgery for intracranial aneurysm clip reconstruction with RVP at the University Bonn Medical Center from June 2015 to February 2021 were reviewed retrospectively. RVP protocol was initiated based on aneurysm size and location. Data of perioperative evaluation and management, including age, sex, weight, relevant medical history, pre-existing neurologic deficits, and co-morbidities, were collected and analyzed.

Intraoperative data including anesthetic technique, number and frequency of rapid ventricular pacing, and hemodynamics including blood pressure, heart rhythm, and heart rate before, during, and after RVP were evaluated. Complications encountered such as missing blood pressure or flow reduction, arrhythmias, and prolonged periods of hypotension were collected. Post-operative data included cardiac and respiratory complications as well as laboratory results, i.e., high-sensitive Troponin T (hsTnT).

Statistical analyses were performed using Prism 8 (GraphPad Software, San Diego, CA, USA). Data are presented as mean ± SEM; results were calculated using unpaired *t*-test with Welch’s correction. Fisher’s exact test was used for sub group analysis to detect non-random association between categorical variables.

### Anesthesia Management

Basic monitoring methods including continuous pulse oximetry, electrocardiogram (ECG), and non-invasive blood pressure were established upon arrival with the anesthesia team. In elective surgery this was in the operation area, in some cases of emergency surgery this was in the emergency department or intervention suite of the neuroradiological department. A peripheral venous catheter was established in every patient if not preexisting.

Anesthesia induction in elective surgery was performed under continuous bispectral index monitoring (BIS; XP-sensor, Covidien plc, Dublin, Ireland) and as total intravenous anesthesia (TIVA). The opioid used was remifentanil (Hameln Pharma GmbH, Hameln, Germany). Propofol (B. Braun Melsungen AG, Melsungen, Germany) was administered with target-controlled infusion (TCI) using designated pumps (Alaris PK, BD, Heidelberg, Germany). Propofol plasma concentrations were calculated based on the pharmacokinetics and pharmacodynamics (PK/PD) using the Schnider model and Propofol infusion was started with a target effect-site drug concentration (Cet) of 3.5 until BIS decreased to the desired range, between 40 and 60 [14]. Subsequently, infusion rate was reduced to keep the patient in the recommended BIS range. Rocuronium (0.6–1 mg/kg body weight; Inresa Arzneimittel GmbH, Freiburg, Germany) was given for orotracheal intubation; successful intubation was controlled by auscultation and capnography.

Anesthesia induction in emergency surgery was carried out as a rapid sequence induction, administering remifentanil followed by a body weight adapted propofol bolus (1.5–2 mg/kg body weight) and rocuronium (1 mg/kg body weight). After orotracheal intubation, balanced anesthesia using sevoflurane was continued during the procedure in the angiographic suite.

All patients received an arterial cannula for continuous invasive blood pressure monitoring after the induction of anesthesia. Both the elective patients, as well as the emergency patients after transferal from the angiographic suite, received an ultrasonic guided central-venous catheter and 6 FR vascular introducer sheath (6 FR × 10 cm; Arrow, Teleflex Medical GmbH, Fellbach, Germany). Subsequently, a bipolar pacing catheter (Edwards Lifesciences Services GmbH, Untereschbach, Germany) was advanced through the venous sheath, by the anesthesiologist, into the right ventricle via trans esophageal echocardiographic guidance. Correct positioning was confirmed under continuous ECG monitoring of the ventricular capturing of the cardiac stimulator (Pace 203 H, OSYPKA AG, Rheinfelden, Germany), inducing an impulse at a low threshold.

In the surgical theatre, patients were positioned as required and the head was fixed in a Mayfield clamp. Aneurysm preparation was performed by a single neurosurgeon (senior author) and executed microsurgically. Intraoperative noninvasive angiography using indocyanin green (ICG; Diagnostic Green GmbH, Aschheim-Dornach, Germany) was performed to further illustrate the details of the aneurysm and vascular connections, as well as the treatment results.

RVP was initiated during the procedure for up to 60 s upon the neurosurgeon’s request. Pacing rates depended on patients’ risk factors and successful pacing as monitored by visual collapse of the aneurysmal sack as well as reduction of blood pressures. Repetitive RVP was performed if required. Use and effect of RVP in clip reconstruction surgery of intracranial aneurysms are exemplified in Appendix A.

Post-operation, all patients were transferred to the intensive care unit (ICU). The pacing catheter was removed, and the introducer sheath remained on admission to ICU.

## 3. Results

In this retrospective analysis, 18 patients were prepared for clipping of intracranial aneurysm using RVP between June 2015 and February 2021. Surgery was elective in nine patients with uIA, whereas it was urgent in eight patients undergoing surgery for rIA. One patient suffering from aneurysmatic SAH was excluded due to delayed hospitalization and presence of cerebral vasospasm necessitating staged treatment.

Patient characteristics did not differ between the groups (Table 1).

Patients in the rIA group did not undergo extended anesthesia or cardiac evaluation. Routine anesthesia evaluation was carried out preoperatively in uIA patients. One patient in the rIA group had a history of coronary artery disease and another a history of paroxysmal atrial fibrillation. All patients received continuous 3-lead ECG monitoring, presented with sinus rhythm, and showed no ST segment changes.

Aneurysm characteristics (Table 2 and Table 3) were similar between groups: mean diameter was 16.7 ± 2.13 mm in the uIA group as compared to 22.5 ± 4.12 mm in the rIA group (*p* = 0.24). Location did not differ between groups as 78% of aneurysms were found in the anterior circulation in the uIA group and 62.5% in the rIA group (*p* = 0.62).

In the rIA group, mean Hunt & Hess grade was 2.9 and mean Fisher grade was 2.9. In one patient from the rIA group surgery was performed in the sitting position, and all other patients were treated in supine position.

Anesthesia induction time was not different between groups and lasted 58.3 ± 3.6 min in the uIA group as compared with 44.8 ± 5.9 min in the rIA group (*p* = 0.08) (see Figure 1). Puncture sites for central venous catheters and introducer sheaths were internal jugular veins in all but two cases. These two patients from the rIA group received subclavian catheters. Complications during anesthesia induction or catheter placement were not reported in either group.

RVP frequencies and cycles were not different between groups (Figure 2 and Table 4). The average number of cycles required was 2.2 in the uIA group as compared to 1.8 in the RIA group (*p* = 0.41). Pacing frequencies ranged from 100–220 per minute (mean 170.9 ± 9.1 per minute) in the uIA group and 130–220 per minute (mean 168.6 ± 7.4 per minute) in the RIA group (*p* = 0.85). Mean arterial pressure (MAP) in the uIA group declined significantly during RVP (*p* < 0.0001), and returned back to pre-RVP values after termination of RVP (*p* = 0.27; see Figure 3). Similarly, in the RIA group, MAP declined significantly during RVP (*p* < 0.0001), and returned to pre-RVP values after RVP was terminated (*p* = 0.18). Comparing MAP between both groups, MAP was significantly lower in the RIA group before, during, and after RVP (Figure 4).

HsTnT values in the uIA group were available for 55% preoperatively and 78% postoperatively. The cut-off for hsTnT in the institution’s laboratory is 14 ng/L. Mean preoperative hsTnT was 6.48 ng/L (range 3.0–14.8 ng/L) as compared with 7.65 ng/L (range 5.2–14.2) postoperatively (*p* = 0.67; Figure 5).

In the rIA group, values were available for 50% preoperatively and 100% postoperatively. Mean preoperative hsTnT was 36.38 ng/L (range 6.6–118 ng/L) as compared with 65.75 ng/L (range 6.4–392 ng/L) postoperatively (*p* = 0.59).

Complications occurred in one patient in the rIA group (*p* = 0.47). This patient presented with pulseless ventricular fibrillation two minutes after successful RVP and required cardiopulmonal resuscitation (CPR). Noteworthy, the onset of SAH of this patient presented as out-of-hospital cardiac arrest with return of spontaneous circulation on admission, and between RVP and CPR during surgery the aneurysm ruptured again. No further arrhythmias or ECG signs of ischemia, i.e., ST segment changes, were observed in either group.

## 4. Discussion

Treatment of intracranial aneurysms can be delicate and challenging, especially for ruptured complex aneurysms. For safe aneurysm treatment, the aneurysm neck, as well as the parent artery, perforators, and involved branches have to be exposed in order to gain proximal perfusion control. However, a comprehensive view of the surgical field is not always possible, especially in large or giant aneurysms when the parent artery, perforators, and branches are located beneath the aneurysm.

Over the past decades, various strategies and techniques have been established for the treatment of complex intracranial aneurysms, including temporary clipping, cardiac standstill during deep hypothermia, as well as adenosine-induced cardiac arrest [15,16,17].

Compared with pharmacological techniques or hypothermia, rapid ventricular pacing has been reported in neurovascular surgery as early as 1971, but has been reintroduced only in 1998 in interventional aortic repair and is time predictable and unaffected by medical conditions such as allergies [18,19]. Since 2011, case reports and two case series have shown the feasibility of RVP for clip reconstruction surgery of unruptured intracranial aneurysm [9,10,11,12,13].

To our knowledge, there are no data available evaluating the use or safety of RVP in aneurysm surgery for the surgical treatment of ruptured cerebral aneurysms in acute SAH. We presented a single-center retrospective analysis comparing the feasibility and safety of RVP in elective clip reconstruction surgery for uIA to emergency clip reconstruction surgery of rIA.

In both groups, no complications occurred regarding anesthesia induction. Specific, establishing central-venous sheaths as well as introduction and positioning of the pacemaker electrodes were uneventful in all patients. Furthermore, duration of anesthesia induction was similar between groups, suggesting that the use of transient pacemakers is a feasible technique for the experienced anesthetist.

The timing of controlled hypotension and return to previous blood pressures is essential for both proper clip placement and adequate brain perfusion. Effective controlled and, most importantly, timed hypotension was accomplished in all patients. In both groups, immediate and significant MAP decline was achieved without prolonged hypotension after RVP was terminated. Patients presenting with SAH often suffer from hemodynamic instability due to excessive catecholamine surge, intracranial hypertension, or neurogenic shock. Surprisingly, RVP in those patients did not result in any hemodynamic instability or prolonged hypotension. Additionally, in this study, all patients in the rIA group had significantly lower MAPs compared with patients from the uIA group, but did not present with more complications including arrhythmias after RVP.

Urgent and emergency procedures constitute increased anesthetic risks since the anesthetist has only reduced information on the patients’ medical history and time for extended diagnostic is limited. In our study, RVP in patients with rIA was executed despite missing cardiac evaluation and did not lead to significant complications. The preoperative cardiac workup in the uIA group was basic compared to Konczalla et al., who performed TTE and stress echocardiography. All patients presented with sinus rhythm, no ST segment changes on continuous 3-lead ECG, and did not develop arrhythmias. Especially in SAH, unspecific increased hsTnT levels as well as ECG changes have been reported and are subsumed under the term of stunned myocardium [20]. HsTnT concentrations were not elevated before surgery, and more importantly did not increase significantly after surgery, suggesting that RVP did not result in detectable myocardial injury. Furthermore, repetitive RVP with up to five consecutive cycles did not lead to any complication. Noteworthy, one patient in the rIA group had known coronary artery disease and exhibited similar hsTnT concentrations before and after RVP during cerebral aneurysm surgery.

Furthermore, one patient was admitted to our hospital after out of hospital cardiac arrest and approximately 15 min CPR with return of spontaneous circulation, exhibiting SAH grade Hunt & Hess 5, Fisher 3. In this patient, anesthesia induction was eventless. With regard to recent CPR, RVP was induced with lower frequencies (130 per minute) than usual and adequate MAP drop was measured. Intraoperatively, the aneurysm ruptured twice despite prompt clip placement. Following the second rupture the patient exhibited ventricular fibrillation (VF) and had to undergo CPR for 4 min before return of spontaneous circulation. In retrospect, it remains unclear if VF was triggered by RVP or due to the SAH itself.

Owing to the retrospective design of this study, there are some limitations. Mainly, the number of patients was little and clear conclusions cannot be drawn. However, aneurysm surgery for SAH necessitating RVP is not a routine procedure, so large study populations are not to be expected. HsTnT values were incomplete. Although the existing data show promising results regarding cardiac safety, hsTnT should be measured regularly to verify this.

## 5. Conclusions

In summary, our data suggest that RVP is feasible even in an urgent setting during surgical reconstruction of complex ruptured intracranial aneurysms in the absence of extended cardiologic evaluation. Missing cardiac workup did not raise periprocedural complications. With the exception of VF after intraoperative aneurysm rupture, no arrhythmias were detected while RVP could be performed effectively. Furthermore, no complications were observed for central venous sheath insertion and induction time was not prolonged.

## Figures and Tables

**Figure 1 jcm-10-05406-f001:**
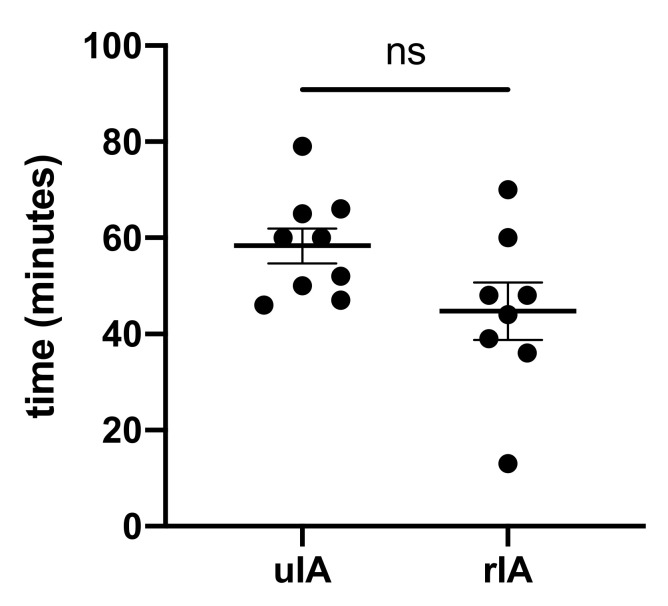
Anesthesia induction time did not differ.

**Figure 2 jcm-10-05406-f002:**
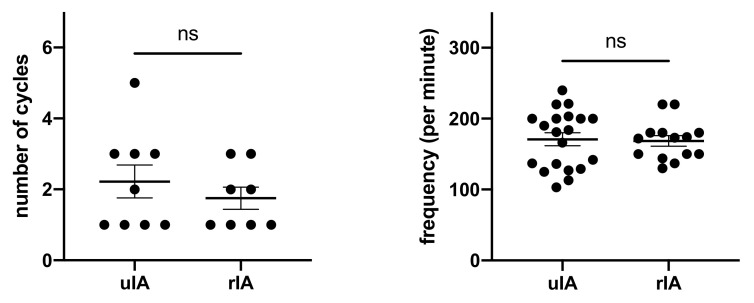
Amount of cycles and frequencies used for RVP were similar.

**Figure 3 jcm-10-05406-f003:**
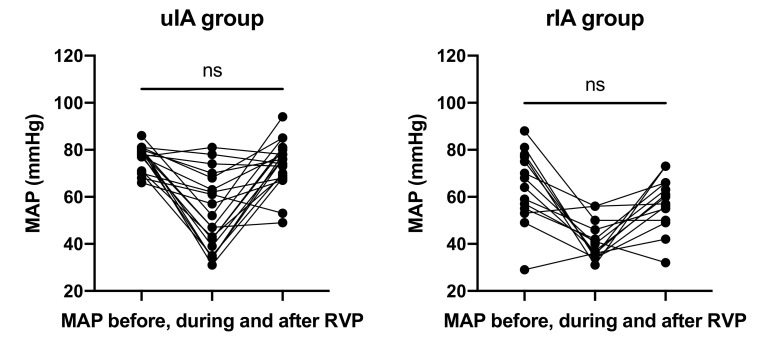
MAP declined significantly during RVP (*p* < 0.0001) and returned to initial values.

**Figure 4 jcm-10-05406-f004:**
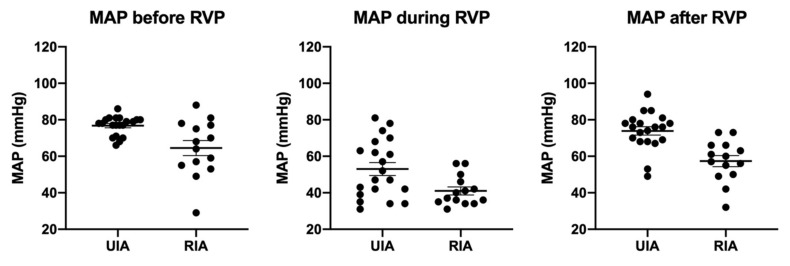
Throughout all phases, MAP was significantly lower in the rIA group.

**Figure 5 jcm-10-05406-f005:**
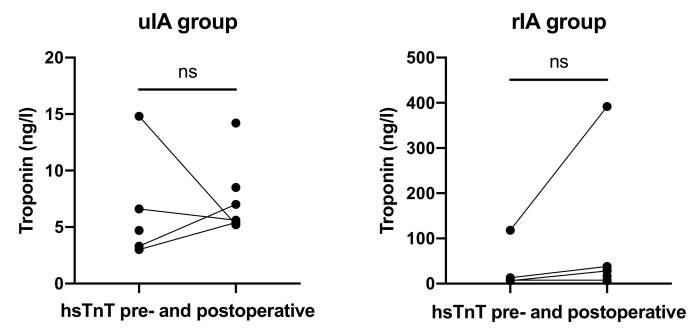
HsTnT levels before and after RVP did not differ in both groups.

**Table 1 jcm-10-05406-t001:** Patient characteristics.

	uIA Group	rIA Group	*p* Value
number	9	8	
mean age	49.2 ± 4.92	56.1 ± 4.03	0.29
sex			0.99
male	3	2	
female	6	6	
mean BMI	25.8 ± 1.96	28.7 ± 2.45	0.37
arterial hypertension	2	3	0.62

**Table 2 jcm-10-05406-t002:** Aneurysm characteristics.

		uIA		
Location	Size (mm)	Hunt & Hess	Fisher	Procedure Related Complications
ICA-Pcom	26	n.a.	n.a.	None
MCA bifurcation	18	n.a.	n.a.	None
Acom	6	n.a.	n.a.	None
ICA-Pcom	16	n.a.	n.a.	None
ICA-bifurcation	20	n.a.	n.a.	None
Paraclinoid ICA	19	n.a.	n.a.	None
Paraclinoid ICA	22	n.a.	n.a.	None
MCA	8	n.a.	n.a.	None
Paraclinoid ICA	22	n.a.	n.a.	None
		** rIA **		
**Location**	** Size (mm) **	** Hunt & Hess **	** Fisher **	** Procedure Related Complications **
ICA backwall	25	5	3	Ventricular fibrillation
MCA	40	3	3	None
ICA-Pcom	27	4	3	None
AICA	6	2	3	None
Paraclinoid ICA	21	2	3	None
ICA-PCom	15	5	3	None
Paraclinoid ICA	11	1	1	None
MCA bifurcation	35	1	4	None

Posterior communicating artery (Pcom), middle cerebral artery (MCA), anterior communicating artery (Acom), internal carotid artery (ACI), Anterior inferior cerebellar artery (AICA).

**Table 3 jcm-10-05406-t003:** Aneurysm differences between groups.

	uIA Group	rIA Group	*p* Value
aneurysm size	16.7 ± 2.13	22.5 ± 4.12	0.24
aneurysm location			0.62
anterior circulation	7	5	
posterior circulation	2	3	
Hunt & Hess grade		2.9 (1–5)	
Fisher grade		2.9 (1–4)	

**Table 4 jcm-10-05406-t004:** Rapid ventricular pacing (RVP) setting and median arterial pressure (MAP) results.

	uIA Group	rIA Group	*p* Value
RVP cycles	2.2	1.8	0.41
RVP frequency	170.9 ± 9.1	168.6 ± 7.4	0.85
MAP before RVP	76.8 ± 1.2	64.5 ± 4.1	0.0119
MAP during RVP	53.0 ± 3.5	41.0 ± 2.2	0.0071
MAP after RVP	73.9 ± 2.3	57.4 ± 3.1	0.0002

## Data Availability

The data presented in this study are available on request from the corresponding author.

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
