# Peer review of "Rapid Ventricular Pacing as a Safe Procedure for Clipping of Complex Ruptured and Unruptured Intracranial Aneurysms"

_jcm, 2021, doi:10.3390/jcm10225406_

Round 1

Reviewer 1 Report

In this study authors present their experience with the use of rapid ventricular pacing to achieve hypotension during clipping of intracranial aneurysms. The technique has been described in neurosurgery with reports in both open and endovascular neurosurgery (Waqas MDossani RHVakharia K, et al Complete flow control using transient concurrent rapid ventricular pacing or intravenous adenosine and afferent arterial balloon occlusion during transvenous embolization of cerebral arteriovenous malformations: case series. 

Author Response

Reviewer 1:

In this study authors present their experience with the use of rapid ventricular pacing to achieve hypotension during clipping of intracranial aneurysms. The technique has been described in neurosurgery with reports in both open and endovascular neurosurgery (Waqas M, Dossani RH, Vakharia K, et al Complete flow control using transient concurrent rapid ventricular pacing or intravenous adenosine and afferent arterial balloon occlusion during transvenous embolization of cerebral arteriovenous malformations: case series. Journal of NeuroInterventional Surgery 2021;13:324-330) and https://doi.org/10.1227/NEU.0b013e318236d84a. 

We would like to thank the Reviewer for his suggestion and have integrated the manuscript in the references

The manuscript shows the safety and feasibility of the technique. The concept is not new however, the quality of the manuscript may be improved by addressing following comments 

1) What was the criteria for selection of patients for RVP. Does the location, size and complexity play a role? Is RVP reserved to control intra-operative rupture or is used uniformly across all cases. 

We have added the following sentence to the manuscript: “RVP protocol was initiated based on aneurysm size and location.”

2) Please clarify if the patients were consecutive. Please provide the praise location of the aneurysm. 

We included table 2 into the manuscript. (die Seite des Aneurysmas würde ich löschen, ist mE nicht relevant).

3) What was the duration of each cycle of RVP?

The duration of the individual cycles was up to 60s. This information was include into the method section. “RVP was initiated during the procedure for up to 60s upon the neurosurgeon’s request. “

4) Who performs the RVP. Is an interventional cardiologist involved. 

We would like to thank the reviewer for pointing out this important point. The entire procedure including the insertion of the stimulation electrode and the RVP has been performed by a trained anesthesiologist in the absence of a cardiologist. We have revised the method section as follow: “Subsequently, a bipolar pacing catheter (Edwards Lifesciences Services GmbH, Un-tereschbach, Germany) was advanced through the venous sheath into the right ventricle via trans esophageal echocardiographic guidance and correct positioning was confirmed under continuous ECG monitoring of the ventricular capturing of the cardiac stimulator (Pace 203 H, OSYPKA AG, Rheinfelden, Germany) induced impulse at a low threshold, performed by the anesthesiologist.” We highlighted that the entire procedure was performed by an anesthesiologist to avoid the statement that this was performed in the absence of an cardiologist and hope that the Reviewer is satisfied with this statement. However, if the Reviewer insists, we will include the absence.

5) Discuss the advantages and disadvantages compared to other techniques to achieve hypotension such as adenosine induced hypotension and hypothermia etc 

We have included the advantages of RVP compared to pharmacological treatments or hypothermia and revised the discussion as followed: “Compared to pharmacological techniques or hypothermia, rapid ventricular pacing that has been reported in neurovascular surgery as early as 1971 but has been reintroduced only in 1998 in interventional aortic repair is time predictable and not affected by medical conditions such as allergies.” Furthermore, intra-aneurysmal blood can be redistributed more easily during the clipping procedure when parent vessels are not blocked due to temporary clipping i.e., facilitating vascular reconstruction especially in large or giant aneurysms. 

We would like to thank the Reviewer for his valuable suggestions and the time he invested into our manuscript. We feel that addressing his comments improved the quality of our submission substantially.

Reviewer 2 Report

I have reviewed the article entitled  “Rapid Ventricular Pacing as a Safe Procedure for Clipping of Complex Ruptured and Unruptured Intracranial Aneurysms”

Authors presented retrospective analysis of 18 (8 ruptured and 9 unruptured, 1 patient unclear)  cerebral aneurysm patients underwent surgical clipping with using rapid ventricular pacing. First of all, I congratulate the authors for their work.

I prefer to read more about the preoperative decision process for RVP and see a table contains preop characteristic, postop complications and surgical results (8 ruptured and 9 unruptured, 1 patient unclear).  

I also believe that a descriptive narration of the supplementary video including preoperative images at the beginning of the video, will increase the quality of the paper. 

I think, reading flow is interrupted by some unnecessary abbreviations: OHCA (used 1 time) and ROSC (used 2 times). These interruptions can be avoided with small changes in the manuscript. I also recommend using “uIA” instead of UIA and “rIA” instead of RIA.

Author Response

I have reviewed the article entitled “Rapid Ventricular Pacing as a Safe Procedure for Clipping of Complex Ruptured and Unruptured Intracranial Aneurysms”

Authors presented retrospective analysis of 18 (8 ruptured and 9 unruptured, 1 patient unclear) cerebral aneurysm patients underwent surgical clipping with using rapid ventricular pacing. First of all, I congratulate the authors for their work.

            We would like to thank the reviewer for his complaisant evaluation!

I prefer to read more about the preoperative decision process for RVP and see a table contains preop characteristic, postop complications and surgical results (8 ruptured and 9 unruptured, 1 patient unclear).

            The following information including the requested information have been added to table 1

uIA

Location

Size (mm)

Hunt & Hess

Fisher

Procedure related complications

Aneurysmocclusion

ICA-Pcom

26

n.a.

n.a.

None

Complete

MCA bifurcation

18

n.a.

n.a.

None

Complete

Acom

6

n.a.

n.a.

None

Complete

ICA-Pcom

16

n.a.

n.a.

None

Complete

ICA-bifurcation

20

n.a.

n.a.

None

Complete

Paraclinoid ICA

19

n.a.

n.a.

None

Complete

Paraclinoid ICA

22

n.a.

n.a.

None

Complete

MCA

8

n.a.

n.a.

None

Complete

Paraclinoid ICA

22

n.a.

n.a.

None

Complete

rIA

Location

Size (mm)

Hunt & Hess

Fisher

Procedure related complications

Aneurysm occlusion

ICA backwall

25

2

3

None

Complete

MCA 

40

3

3

None

Complete

ICA-Pcom

27

4

3

None

Complete

AICA

6

2

3

None

Complete

Paraclinoid ICA

21

2

3

None

Complete

ICA-Pcom

15

5

3

Ventricular fibrilation

Complete

Paraclinoid ICA

11

1

1

None

Complete

MCA bifurcation

35

1

4

None

Complete

posterior communicating artery (Pcom), middle cerebral artery (MCA), anterior communicating artery (Acom), internal carotid artery (ACI), Anterior inferior cerebellar artery (AICA), internal carotid artery (ICA).

I also believe that a descriptive narration of the supplementary video including preoperative images at the beginning of the video, will increase the quality of the paper. 

The reviewer is pointing out an important fact. We have revised the video and included preoperative images at the beginning as well as a brief text describing the images and hope that these information will help the reader understanding the impact of the images. “A 47 year old male patient was diagnosed with an unruptured, partially thrombosed large (20mm) aneurysm of the ICA bifurcation. The following sequence shows the diagnostic findings, and rapid ventricular pacing during cerebral aneurysm treatment.”

I think, reading flow is interrupted by some unnecessary abbreviations: OHCA (used 1 time) and ROSC (used 2 times). These interruptions can be avoided with small changes in the manuscript. I also recommend using “uIA” instead of UIA and “rIA” instead of RIA.

Abbreviations for out of hospital cardiac arrest and return of spontaneous circulation have been erased. uIA and rIA have been changed according to reviewers’ suggestion.

We would like to thank the reviewer for the time he invested into our submission and his valuable comments. We feel that addressing the reviewers suggestions, and in particular the revision of the video, substantially improved the quality of our manuscript.